Application of an interpretable machine learning model based on optimal feature selection for predicting triple-vessel coronary disease: a multicenter retrospective study

Hou Ling 1
He Ke 1
Zhao Jinbo 2
Su Ke 2
Zhang Changjiang zcj2008@163.com 1
1 Minda Hospital of Hubei Minzu University , Enshi , China
2 Central Hospital of Tujia and Miao Autonomous Prefecture , Enshi , China
Nogoy Nicole
Electronic publication date: 2025 Dec 9
Publication date: 2025
Volume: 13
Electronic Location ID: e20423
Received 2025 Jul 22; Accepted 2025 Oct 29
Copyright: ©2025 Hou et al.
Copyright year: 2025
Copyright holder: Hou et al.
License: This is an open access article distributed under the terms of the Creative Commons Attribution License, which permits unrestricted use, distribution, reproduction and adaptation in any medium and for any purpose provided that it is properly attributed. For attribution, the original author(s), title, publication source (PeerJ) and either DOI or URL of the article must be cited.
License URL: https://creativecommons.org/licenses/by/4.0/

Keywords: Triple-vessel disease, Remnant cholesterol-inflammatory index, Feature selection, Machine learning, Risk prediction

Funding: Natural Science Foundation of Hubei Province JCZRYB202501509 This work was supported by the Natural Science Foundation of Hubei Province (No. JCZRYB202501509). The funders had no role in study design, data collection and analysis, decision to publish, or preparation of the manuscript.

==============================
Objective

This study aimed to evaluate the predictive value of the remnant cholesterol-inflammatory index (RCII) in assessing the risk of triple-vessel disease (TVD), and to construct a comparative framework of predictive models using six machine learning algorithms based on RCII and other clinical features for identifying high-risk individuals.

Methods

In this retrospective multicenter study, we enrolled 2,911 patients who underwent coronary angiography between January 1, 2024, and December 31, 2024, at two tertiary hospitals. Clinical and laboratory data were collected. Feature selection was performed using both Least Absolute Shrinkage and Selection Operator (LASSO) regression and multivariate logistic regression. Six machine learning (ML) algorithms were trained for risk prediction, with multilayer perceptron (MLP) selected as the optimal model for the final feature set. Model performance was assessed using the area under the receiver operating characteristic curve (AUC), positive predictive value (PPV), and F1 score. SHapley Additive exPlanations (SHAP) analysis was applied to interpret feature contributions and interactions.

Results

A total of 16 features were selected by LASSO regression, while multivariate logistic regression identified six independent predictors. Four overlapping features—gender, age, aspartate aminotransferase (AST), and RCII—were used for ML model development. Among the six models, the MLP demonstrated the best overall performance on the test set. SHAP analysis revealed that RCII, age, AST, and gender were the top contributors to model prediction, with RCII showing notable interaction effects with other variables, highlighting its both independent and synergistic role in TVD risk stratification.

Conclusion

RCII, as a composite biomarker integrating lipid metabolism and chronic inflammation, demonstrates strong predictive utility in identifying individuals at high risk for triple-vessel coronary disease.

Background

Coronary artery disease (CAD), driven primarily by atherosclerosis, remains one of the leading causes of death and disability worldwide, posing a substantial public health burden (Khan et al., 2020). According to the latest Global Burden of Disease (GBD) data, the mortality rate associated with CAD continues to rank among the highest globally, making it a major threat to human health (Zhang et al., 2025). In China, posing a significant challenge to national healthcare systems. The large number of affected individuals underscores the urgent need for precise risk stratification and effective clinical management of coronary lesions.

Atherosclerosis is a complex, chronic, and progressive pathological process, often characterized by heterogeneous plaque distribution involving multiple vascular territories and segments (Jiang et al., 2025; Xing & Lin, 2025). As the disease advances, the accumulation of atherosclerotic plaque and the degree of luminal narrowing progressively worsen, evolving from mild single-vessel stenosis to severe multi-vessel obstruction (Marino et al., 2019). Among the various forms of multi-vessel disease (MVD), triple-vessel disease (TVD)—defined by stenosis (≥50%) in all three major coronary arteries—is considered the most severe subtype, associated with greater ischemic burden, more complex therapeutic decision-making, and a markedly worse prognosis compared to single- or double-vessel disease (Bittl et al., 2016; Giustino et al., 2015; Ng & Bax, 2020). Early identification and risk prediction of TVD are thus of paramount clinical importance.

Emerging evidence has underscored the pivotal roles of dyslipidemia and chronic low-grade inflammation in the initiation and progression of atherosclerotic lesions (Chevalier et al., 2025; Ramos et al., 2025). Remnant cholesterol (RC), the cholesterol content of triglyceride-rich lipoproteins excluding LDL-C, has been shown to promote atherosclerosis and cardiovascular events through mechanisms involving lipid deposition, oxidative stress, and inflammatory activation (Doi, Langsted & Nordestgaard, 2025; Nordestgaard et al., 2025). Mechanistically, RC can penetrate the endothelium and be taken up by macrophages to form foam cells, while simultaneously inducing endothelial dysfunction and oxidative stress, thereby accelerating plaque formation. Concurrently, high-sensitivity C-reactive protein (hsCRP), a well-established inflammatory biomarker, is independently associated with elevated risks of CAD, heart failure, and stroke (Arnold & Koenig, 2025; Mehta, De Goma & Shapiro, 2024; Pan et al., 2025). HsCRP contributes to atherogenesis by promoting endothelial adhesion molecule expression, facilitating leukocyte recruitment, and activating the monocyte–macrophage axis. Clinical studies suggest that disturbances in lipid metabolism and systemic inflammation frequently co-occur in CAD patients and act synergistically to exacerbate overall atherosclerotic disease burden (Choi et al., 2024; Fromentin et al., 2022).

To integrate these two key pathological dimensions, the remnant cholesterol-inflammatory index (RCII) has recently been proposed as a composite biomarker reflecting both lipid and inflammatory status. Although preliminary studies have shown an association between RCII and adverse cardiovascular events (Yu et al., 2025b), its utility in predicting more complex forms of coronary disease such as TVD remains insufficiently explored. Importantly, this study is the first to apply RCII within a machine learning framework for the prediction of triple-vessel disease. Accordingly, the present work aims to systematically evaluate the predictive value of RCII for TVD, with the ultimate goal of improving early risk assessment and informing clinical decision-making in patients with suspected or established CAD.

Methods

Study population

This retrospective observational study enrolled patients who were hospitalized and underwent coronary angiography between January 1 and December 31, 2024, at two medical centers in China: Enshi Tujia and Miao Autonomous Prefecture Central Hospital and Minda Hospital of Hubei Minzu University. Eligible participants were those diagnosed for the first time with TVD, defined as ≥50% stenosis in all three major coronary arteries—the left anterior descending, left circumflex, and right coronary arteries—based on coronary angiographic findings (Li et al., 2024). Patients were excluded if they had known structural or functional cardiac disorders, malignancies, or incomplete key biochemical data, particularly levels of high-sensitivity hsCRP and the full lipid profile (total cholesterol, triglycerides, HDL-C, LDL-C). The study was conducted in accordance with the Declaration of Helsinki and approved by the institutional ethics committees of Enshi Tujia and Miao Autonomous Prefecture Central Hospital (Approval No. 202500201). As a retrospective study, the requirement for informed consent was waived, and all patient data were anonymized to protect privacy. A total of 2,911 patients meeting the inclusion criteria were ultimately included analyzed (Fig. 1). Patient data were accessed and extracted from the hospital information systems between March 1 and March 15, 2025.

Figure 1 Flowchart of patient enrollment and inclusion criteria.

Data collection

Clinical and laboratory data were systematically collected from electronic medical records. Demographic information included age and sex. Medical history included hypertension, type 2 diabetes mellitus (T2DM), hyperlipidemia, hyperuricemia, atrial fibrillation, and stroke. Lifestyle data included smoking status. Vital signs included diastolic blood pressure (DBP) and heart rate. Laboratory parameters encompassed: leukocytes, lymphocytes, monocytes, hemoglobin, platelets, hsCRP, aspartate aminotransferase (AST), total bilirubin, direct bilirubin, albumin, total cholesterol, triglycerides, LDL-C, HDL-C, apolipoprotein A1 (LP_A1), apolipoprotein B (LP_B), serum creatinine, fasting glucose, uric acid, and serum potassium. In addition, the RCII was calculated to represent the composite burden of lipid metabolism and inflammation. RC was estimated as: RC = total cholesterol − HDL-C − LDL-C, and RCII was defined as: RCII = (RC × hsCRP)/10 (Chen et al., 2025).

Feature selection

To enhance model performance and reduce noise from redundant variables, two complementary approaches were used for feature selection. First, the least absolute shrinkage and selection operator (LASSO) regression was applied to identify predictive variables. Subsequently, univariate logistic regression was performed, and variables with a significance level of p < 0.05 were further analyzed using multivariate logistic regression to identify independent predictors. Finally, variables selected from both LASSO and multivariate models were compared and integrated to define a stable feature set, which was then used for machine learning model development. Only overlapping predictors were retained to ensure the robustness and interpretability of the ML models.

Model construction, evaluation, and interpretability

Based on the selected features, six machine learning (ML) algorithms were employed to develop prediction models: multilayer perceptron (MLP), decision tree (DT), random forest (RF), k-nearest neighbor (KNN), light gradient boosting machine (LightGBM), and extreme gradient boosting (XGBoost). All data were randomly divided into training and testing sets in a 7:3 ratio. Hyperparameter tuning for all machine learning models was performed within the training set using Bayesian optimization. For each model, relevant hyperparameter search spaces were defined, and five-fold cross-validation was applied during tuning. Model performance metrics were collected at each iteration to identify the optimal hyperparameter combination. Model performance was assessed using the testing set, based on the area under the receiver operating characteristic curve (AUC), overall accuracy, sensitivity, specificity, F1 score, and positive and negative predictive values. The best-performing model was selected for interpretability analysis. To interpret the model outputs, the SHapley Additive exPlanations (SHAP) framework was utilized. SHAP summary plots (scatter and bar plots) were used to evaluate the global contribution and directional effect of each feature on model predictions. SHAP dependence plots were employed to visualize how SHAP values change with feature values, reflecting their marginal effects. Moreover, SHAP interaction plots were generated to investigate potential synergistic effects between RCII and other key variables, capturing complex nonlinear relationships affecting TVD risk. For reproducibility, SHAP analyses were performed with 1,000 background samples and 100 iterations per sample, and interaction values were normalized. All ML modeling and SHAP analyses were conducted using Python (version 3.9) with scikit-learn (version 1.2.2), XGBoost (version 1.7.6), LightGBM (version 3.3.5), and shap (version 0.47.2) packages.

Statistical analysis

All statistical analyses were conducted using Python (version 3.9) and R (version 4.3.2). Continuous variables with a normal distribution were presented as mean ± standard deviation (SD) and compared using the independent sample t-test. Non-normally distributed variables were expressed as medians with interquartile ranges (IQR) and compared using the Mann–Whitney U test. Categorical variables were summarized as counts and percentages and compared using the chi-squared (χ2) test or Fisher’s exact test where appropriate. All statistical tests were two-tailed, and a p-value <0.05 was considered statistically significant.

Results

Baseline characteristics

A comparative analysis of clinical characteristics between the TVD and non-TVD groups is presented in Table 1. The proportion of males was significantly higher in the TVD group. Additionally, the prevalence of hypertension, T2DM, hyperuricemia, and a history of smoking was markedly elevated among patients with TVD. Patients in the TVD group were older and had slightly higher diastolic blood pressure. In terms of inflammatory and hematological parameters, levels of leukocytes and monocytes were significantly increased in the TVD group. Regarding liver function, AST was notably higher in the TVD group. Among metabolic indices, fasting glucose, serum creatinine, uric acid, and potassium levels were all significantly elevated in the TVD group compared to the non-TVD group. Importantly, the RCII was significantly higher in patients with TVD. No significant differences were observed between the two groups in terms of heart rate, hyperlipidemia, atrial fibrillation, stroke history, lymphocyte count, hemoglobin, platelet count, triglycerides, apolipoprotein A1, apolipoprotein B, total bilirubin, direct bilirubin, or serum albumin.

Table 1 Baseline characteristics.

Variables	Non TVB (n = 2,151)	TVB (n = 760)	P value	
Gender=male (%)	1,302 (60.5)	530 (69.7)	<0.001*	
Hypertension=yes (%)	1,228 (57.1)	485 (63.8)	0.001*	
Type 2 Diabetes Mellitus=yes (%)	536 (24.9)	247 (32.5)	<0.001*	
Hyperlipidemia=yes (%)	399 (18.5)	144 (18.9)	0.809	
Hyperuricemia=yes (%)	123 ( 5.7)	64 ( 8.4)	0.009*	
Atrial_Fibrillation=yes (%)	108 ( 5.0)	45 ( 5.9)	0.339	
Stroke=yes (%)	545 (25.3)	205 (27.0)	0.375	
Smoking_history=yes (%)	939 (43.7)	398 (52.4)	<0.001*	
Age (median [IQR])	65.00 [56.00, 72.00]	67.00 [59.00, 73.00]	<0.001*	
DBP (median [IQR])	80.00 [70.00, 87.00]	80.00 [70.75, 90.00]	0.004*	
HR (median [IQR])	72.00 [68.00, 79.00]	72.00 [66.00, 80.00]	0.257	
Leukocytes (median [IQR])	6.17 [5.03, 7.56]	6.48 [5.38, 8.12]	<0.001*	
Lymphocytes (median [IQR])	1.43 [1.10, 1.85]	1.42 [1.04, 1.85]	0.227	
Monocytes (median [IQR])	0.37 [0.29, 0.47]	0.39 [0.31, 0.51]	<0.001*	
Hemoglobin (median [IQR])	137.00 [126.00, 150.00]	138.00 [124.00, 148.25]	0.826	
Platelets (median [IQR])	200.00 [165.00, 240.00]	199.50 [164.00, 235.00]	0.666	
AST (median [IQR])	21.00 [17.00, 29.00]	23.00 [17.00, 34.00]	<0.001*	
Total_bilirubin (median [IQR])	11.10 [8.50, 14.70]	11.50 [8.78, 15.40]	0.121	
Direct_bilirubin (median [IQR])	2.80 [1.90, 3.90]	2.80 [1.90, 3.90]	0.839	
Albumin (median [IQR])	39.00 [36.20, 42.39]	38.80 [35.71, 42.15]	0.241	
Triglycerides (median [IQR])	1.32 [0.96, 1.88]	1.37 [0.99, 1.97]	0.055	
RCII (median [IQR])	0.06 [0.02, 0.16]	0.07 [0.02, 0.24]	0.014*	
Apo_A1 (median [IQR])	1.22 [1.02, 1.46]	1.21 [1.02, 1.43]	0.439	
Apo_B (median [IQR])	0.82 [0.64, 1.03]	0.84 [0.65, 1.05]	0.073	
Creatinine (median [IQR])	72.60 [60.15, 87.75]	75.45 [62.20, 89.03]	0.002*	
Glucose (median [IQR])	5.68 [4.86, 7.32]	6.12 [5.13, 8.16]	<0.001*	
Uric_acid (median [IQR])	339.63 [280.65, 406.40]	348.80 [290.55, 416.25]	0.035*	
Potassium (median [IQR])	3.96 [3.73, 4.20]	4.01 [3.75, 4.27]	0.004*	
Notes.

* P < 0.05.

DBP diastolic blood pressure

HR heart rate

AST aspartate aminotransferase

RCII remnant cholesterol-inflammatory index

Feature selection

To optimize model stability and reduce multicollinearity, LASSO regression was first employed. This regularization technique enabled efficient variable shrinkage and selection, ultimately retaining 16 candidate predictors: gender, hypertension, T2DM, hyperuricemia, smoking history, age, leukocytes, lymphocytes, monocytes, hemoglobin, platelets, AST, RCII, LP_B, creatinine, and glucose. In parallel, a traditional statistical approach involving univariate screening followed by multivariate logistic regression identified six independent predictors: Gender, Age, DBP, AST, RCII, and Potassium (Table 2). A comparison of both methods (Fig. 2) revealed four overlapping variables—Gender, Age, AST, and RCII—demonstrating consistent predictive performance across both techniques. These four features were thus selected for ML modeling to construct a more robust and interpretable TVD prediction framework. This conservative strategy was adopted to enhance model stability and interpretability by focusing on predictors consistently identified by both LASSO and logistic regression.

Table 2 Multivariate logistic regression.

Variables	OR	95%CI	p	
Gender	1.323	1.051, 1.665	0.017*	
Hypertension	1.231	0.988, 1.535	0.064	
Type 2 Diabetes Mellitus	1.129	0.895, 1.425	0.305	
Hyperuricemia	1.277	0.853, 1.912	0.235	
Smoking_history	1.165	0.939, 1.445	0.165	
Age	1.019	1.010, 1.029	<0.001*	
DBP	1.009	1.001, 1.017	0.036*	
HR	1.003	0.994, 1.011	0.518	
Leukocytes	1.013	0.964, 1.064	0.618	
Monocytes	1.924	0.974, 3.799	0.059	
AST	1.002	1.000, 1.004	0.03*	
RCII	1.161	1.060, 1.273	0.001*	
Creatinine	1	0.998,1.002	0.83	
Glucose	1.016	0.990, 1.043	0.222	
Uric_acid	1	0.999, 1.001	1	
Potassium	1.514	1.162, 1.973	0.002*	
Notes.

* P < 0.05.

DBP diastolic blood pressure

HR heart rate

AST aspartate aminotransferase

RCII remnant cholesterol-inflammatory index

Figure 2 Feature selection and model tuning using LASSO regression.

(A) Mean squared error (MSE) across different alpha values in the LASSO regression model, showing the optimal regularization parameters (λmin and λ1se). The curve highlights where the model reaches minimum prediction error and where a more parsimonious model is selected. (B) Network visualization of features selected by LASSO regression and logistic regression. Orange nodes represent features selected only by LASSO, purple nodes represent features selected only by logistic regression, and green nodes represent features selected by both methods, illustrating areas of overlap and divergence between the two approaches.

Model comparison and interpretability

The four selected features were used to train six machine learning models: MLP, DT, RF, KNN, LightGBM, and XGBoost. Models were evaluated on both training and test datasets. As shown in Table 3, the RF model achieved the highest performance in the training set, demonstrating superior accuracy and AUC. However, it demonstrated substantial overfitting, as evidenced by performance degradation in the test set. In contrast, the MLP model yielded the best generalization performance on the test set, followed closely by LightGBM. As illustrated in Figs. 3 and 4, the MLP model demonstrated both high stability and robust predictive capacity, and was therefore selected for subsequent interpretability analyses using the SHAP framework.

Table 3 The comprehensive evaluation of the performance of predictive models.

	AUC	Accuracy	Sensitive	F1 score	PPV	NPV	
			Training model				
RF	0.8426011	0.716	0.908	0.625	0.477	0.952	
DT	0.6982215	0.683	0.577	0.488	0.422	0.828	
xgboost	0.6516212	0.591	0.652	0.454	0.349	0.822	
Lightgbm	0.6347239	0.557	0.682	0.446	0.331	0.82	
MLP	0.598234	0.552	0.637	0.426	0.32	0.803	
KNN	0.6486442	0.628	0.57	0.445	0.365	0.81	
			Testing model				
RF	0.5899938	0.539	0.61	0.408	0.307	0.789	
DT	0.5401017	0.603	0.382	0.334	0.297	0.757	
xgboost	0.5735226	0.554	0.548	0.391	0.303	0.777	
Lightgbm	0.5917862	0.541	0.61	0.409	0.308	0.79	
MLP	0.6078635	0.561	0.61	0.42	0.32	0.798	
KNN	0.5996212	0.601	0.518	0.403	0.331	0.787	
Notes.

Abbreviations PPV positive predictive value

NPV negative predictive value

RF random forest

DT decision tree

MLP multilayer perceptron

KNN k-nearest neighbors

Figure 3 Parallel coordinate plots of model performance across six machine-learning (ML) algorithms.

Six ML algorithms were used to develop prediction models: multilayer perceptron (MLP), decision tree (DT), random forest (RF), k-nearest neighbor (KNN), light gradient boosting machine (LightGBM), and extreme gradient boosting (XGBoost). (A) Parallel coordinate plot for the training set displaying multiple performance metrics of the six ML algorithms, allowing comparison of model behavior during training. (B) Parallel coordinate plot for the validation set illustrating model performance on unseen data, highlighting generalization ability and differences among the six algorithms.

Figure 4 Calibration curves of the six machine-learning (ML) models. Prediction models were constructed using six ML algorithms: MLP, DT, RF, KNN, LightGBM, and XGBoost.

(A) Calibration curves for the training set showing the agreement between predicted probabilities and observed outcomes for each model. Curves closer to the diagonal line indicate better calibration. (B) Calibration curves for the validation set evaluating calibration performance on unseen data, reflecting the consistency and reliability of probability predictions across the six ML algorithms.

To elucidate the inner workings of the MLP model and the contribution of each input variable, SHAP analysis was performed. As shown in Fig. 5, SHAP summary bar and scatter plots revealed the relative importance of the four input features in predicting TVD. Notably, RCII ranked third among all features, with its SHAP value showing a strong positive correlation with its actual value. This indicates that elevated RCII significantly increases the predicted risk of TVD. Furthermore, older age, elevated AST levels (indicative of hepatic dysfunction or metabolic stress), and male sex were also associated with higher predicted TVD risk.

Figure 5 SHAP analysis of the logistic regression model.

(A) SHAP summary plot showing the contribution of individual features to model predictions. Each point represents a sample, with color indicating the feature value (blue: low, red: high). Horizontal position reflects the direction and magnitude of SHAP impact. (B) Mean absolute SHAP values ranking global feature importance. Higher bars represent features with greater overall influence on the model’s predictions.

Figure 6 presents SHAP dependence plots, illustrating the quantitative relationships between feature values and model outputs. Both age and RCII exhibited a monotonic increase in SHAP values, particularly when RCII exceeded 0.2, reinforcing its role as a high-risk marker for advanced coronary atherosclerosis. The influence of AST and gender on SHAP values was also evident, with male patients demonstrating consistently higher risk contributions. Finally, Fig. 7 displays the SHAP-based interaction heatmap, highlighting synergistic effects among the four features. Significant interactions were observed between RCII and age, AST, and gender, suggesting that the predictive value of RCII is enhanced when combined with other clinical risk factors.

Figure 6 SHAP dependence plots showing the marginal effects of key features on TVD risk.

Figure 7 Partial dependence contour plots of key predictors.

(A) Contour plot showing the interaction between Gender and RCII and their combined influence on predicted probability. (B) Contour plot showing the interaction between Age and RCII, illustrating how age modifies the effect of RCII on model predictions. (C) Contour plot demonstrating the joint influence of AST and RCII on predicted outcomes.

Discussion

Our findings revealed that patients with TVD were generally older, more likely to be male, and had a significantly higher prevalence of hypertension, T2DM, hyperuricemia, a history of smoking. Inflammatory markers and liver function indicators—particularly AST—were also elevated in the TVD group, alongside notable differences in metabolic profiles. Notably, the RCII was significantly higher in the TVD group and emerged as a robust predictor of TVD, selected by both LASSO regression and multivariate logistic analysis. Interpretability analysis using SHAP further confirmed RCII as a strong positive contributor to the predictive model, with higher RCII values corresponding to an increased risk of TVD. Moreover, significant interactions were observed between RCII and age, AST, and sex, indicating a synergistic effect of dysregulated lipid metabolism and chronic inflammation in the progression of complex coronary lesions. These findings are consistent across multiple analytical approaches and are supported by rigorous feature selection and interpretability analyses, thereby providing confidence in the robustness of the results.

RCII was first proposed by Chen et al. (2025) in 2024 as a composite marker integrating lipid and inflammatory status for predicting stroke risk in older adults. Subsequent studies have supported its association not only with cerebrovascular outcomes but also with all-cause, cardiovascular, and cancer-related mortality (Wang et al., 2025; Yu et al., 2025b). RCII combines RC and hs-CRP, two critical mediators of atherosclerosis that reflect lipid deposition and chronic low-grade inflammation, respectively (Elías-López et al., 2024; Shen & Zhang, 2025). RC, a cholesterol component of triglyceride-rich lipoprotein remnants, can penetrate the endothelium and be phagocytosed by macrophages to form foam cells, thereby accelerating plaque formation (Gomez-Delgado et al., 2024). Elevated RC levels are also associated with endothelial dysfunction and oxidative stress, which promote vascular inflammation (Baratta et al., 2023). hs-CRP, a sensitive marker of systemic inflammation, enhances endothelial expression of adhesion molecules, facilitates leukocyte recruitment, and activates the monocyte–macrophage axis, aggravating local inflammatory responses (Ridker et al., 2017; Yu et al., 2025a). It may also destabilize plaques by upregulating matrix metalloproteinases (Jia et al., 2023; Kumar et al., 2025).

As a combined index, RCII amplifies the individual pathogenic effects of RC and hs-CRP and provides a more comprehensive reflection of the pathophysiological burden of atherosclerosis. While other emerging biomarkers such as lipoprotein (a) have been proposed for CAD risk prediction, they were not found to be significant in our cohort, possibly due to their weaker relevance to complex coronary phenotypes like TVD or limited variability in our population. This further underscores the rationale for focusing on RCII, which integrates both lipid and inflammatory pathways and demonstrated consistent predictive value across multiple analytical approaches. Importantly, the rationale for using RCII in our study is strongly supported by prior evidence, reinforcing that our application of this composite marker to TVD prediction is well-grounded and scientifically justified.

This study is the first to explore the predictive utility of RCII for triple-vessel coronary disease. Compared to individual lipid or inflammatory markers, RCII combines lipid and inflammatory markers, improving predictive granularity by capturing both metabolic and inflammatory pathways involved in atherogenesis. Our results are in line with previous studies that highlight advanced age, male sex, and hepatic/metabolic dysfunction as key contributors to adverse cardiovascular outcomes, further emphasizing the clinical significance of RCII in high-risk coronary phenotypes. From a clinical perspective, RCII thresholds could be applied to guide triage and risk stratification—for example, patients with RCII values above of the studied population may be considered high risk for TVD and prioritized for advanced diagnostic imaging or intensive preventive therapies. Furthermore, integrating RCII into electronic health record systems or cardiovascular risk calculators could enable automated alerts for clinicians, supporting individualized treatment decisions. By providing a quantitative, interpretable measure of combined lipid and inflammatory burden, RCII may help bridge the gap between pathophysiological insight and actionable clinical guidance, facilitating early intervention in high-risk patients.

Limitations

Despite the promising findings, several limitations should be acknowledged. First, the retrospective design and reliance on previously recorded clinical data may introduce potential biases and residual confounding, including the influence of unmeasured factors such as medication use. Second, although the study included a multicenter cohort, it was still regionally restricted, and external validation in independent populations is warranted. Third, our feature selection strategy prioritized only overlapping predictors from LASSO and logistic regression, which enhances interpretability but may have excluded some clinically relevant variables, thereby potentially affecting model robustness. Furthermore, our findings are specific to patients with triple-vessel coronary disease, and the applicability of RCII to other forms or stages of coronary artery disease remains to be determined. Taken together, these considerations highlight the need for further studies to explore these aspects and to confirm the predictive utility of RCII across broader and more diverse CAD populations.

Conclusion

This study is the first to incorporate RCII into a machine learning-based model for predicting the risk of TVD. The findings demonstrate that RCII—which reflects both lipid dysregulation and inflammatory status—is a valuable predictor for identifying high-risk individuals. However, given the retrospective nature of the study, potential biases cannot be entirely excluded. While RCII has been validated in large population-based studies for outcomes such as mortality and stroke, its specific role in CAD is still less well established. Its components—remnant cholesterol and systemic inflammation—are independently linked to CAD risk and severity, providing a strong rationale for its application in CAD prediction. Future prospective, multicenter studies are warranted to validate the clinical utility of RCII in early risk stratification and precision management of complex coronary artery disease.

Supplemental Information

Supplemental Information 1 Code

Supplemental Information 2 Data

Supplemental Information 3 Translation codebook for non-English text in code files

Supplemental Information 4 STROBE checklist

Additional Information and Declarations

Competing Interests

Author Contributions

Human Ethics

Data Availability

The authors declare there are no competing interests.

Ling Hou conceived and designed the experiments, performed the experiments, analyzed the data, prepared figures and/or tables, authored or reviewed drafts of the article, and approved the final draft.

Ke He conceived and designed the experiments, performed the experiments, authored or reviewed drafts of the article, and approved the final draft.

Jinbo Zhao performed the experiments, prepared figures and/or tables, authored or reviewed drafts of the article, and approved the final draft.

Ke Su performed the experiments, prepared figures and/or tables, authored or reviewed drafts of the article, and approved the final draft.

Changjiang Zhang conceived and designed the experiments, analyzed the data, authored or reviewed drafts of the article, and approved the final draft.

The following information was supplied relating to ethical approvals (i.e., approving body and any reference numbers):

The study was conducted in accordance with the Declaration of Helsinki and approved by the institutional ethics committees of Enshi Tujia and Miao Autonomous Prefecture Central Hospital (Approval No. 202500201).

The following information was supplied regarding data availability:

The raw data is available in the Supplemental Files.

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
