# Peer review of "Application of an interpretable machine learning model based on optimal feature selection for predicting triple-vessel coronary disease: a multicenter retrospective study"

_PeerJ, doi:10.7717/peerj.20423_

## Round 0.1 · original submission · Major Revisions

· Academic Editor

Major Revisions

**Language Note:** When preparing your next revision, please ensure that your manuscript is reviewed either by a colleague who is proficient in English and familiar with the subject matter, or by a professional editing service. PeerJ offers language editing services; if you are interested, you may contact us at [email protected] for pricing details. Kindly include your manuscript number and title in your inquiry. – PeerJ Staff

·

Basic reporting

[1] The manuscript is written in professional English, with generally clear and unambiguous phrasing. Minor improvements could enhance precision and readability:
Line 20: “multi-model machine learning framework” could be clarified as “ensemble or comparative ML framework.”
Line 34: “shows strong potential” is vague; consider quantifying predictive strength.

[2] The background is well contextualized, citing relevant epidemiological and mechanistic literature. The introduction effectively frames the knowledge gap regarding RCII and TVD. References are current and appropriate.

[3] The manuscript follows standard IMRaD format. Figures and tables are relevant, well-labeled, and appropriately referenced. Raw data is summarized in tables, and SHAP plots are described in detail. However:
The editorial team may need to check if all figures (e.g., SHAP plots, calibration curves) are of sufficient resolution and include axis labels and legends.
Table 3 could benefit from clearer formatting to distinguish training vs. test performance.

[4] The manuscript states that data is available upon reasonable request. For PeerJ compliance, consider depositing anonymized datasets in a public repository (e.g., Dryad, Figshare) with metadata.

[5] The study is self-contained and presents a coherent unit of publication. No evidence of salami slicing.

Experimental design

[1] The research question is well-defined and clinically meaningful. The study addresses a clear gap in predictive modeling for TVD using RCII.

[2] Ethical approval is documented (Approval No. 202500201), and patient data is anonymized. The retrospective design is appropriate for the study's aims.

[3] Methodology is described in sufficient detail to allow replication. Feature selection via LASSO and logistic regression is standard and well-justified. ML model construction and SHAP interpretability are clearly explained.

[4] Clarify hyperparameter tuning procedures for ML models.

[5] Specify software packages used for SHAP analysis.

Validity of the findings

[1] The study does not claim novelty as a criterion but offers meaningful replication and extension of RCII utility. The rationale for using RCII is well-supported.

[2] The dataset is large (n=2,911), multicenter, and well-characterized. However, model performance metrics suggest modest predictive power:
MLP test AUC = 0.608, F1 = 0.42
RF shows overfitting (train AUC = 0.84 vs test AUC = 0.59)
Consider:
External validation on an independent cohort.
Inclusion of calibration metrics (e.g., Brier score).

[3] Conclusions are appropriately linked to results. RCII is positioned as a promising biomarker, but claims are tempered by limitations. The discussion acknowledges retrospective design and the lack of external validation.

Additional comments

1. Language and Expression
Issues:
• Some phrasing is vague or overly general.
• Minor grammatical inconsistencies and technical imprecision.
Suggestions:
• Abstract (Line 34): Replace “shows strong potential” with a quantified statement, e.g., “demonstrated an AUC of 0.61 in predicting TVD.”
• Methods (Line 20): Clarify “multi-model machine learning framework” as “comparative modeling using six ML algorithms.”
• Discussion (Lines 199–200): Avoid subjective phrases like “offers a more integrated risk measure”; instead, state “RCII combines lipid and inflammatory markers, improving predictive granularity.”

2. Literature and Contextual Framing
Issues:
• RCII is introduced with limited mechanistic depth early in the manuscript.
• The novelty of applying RCII to TVD is underemphasized.
Suggestions:
• Expand the introduction to include a brief mechanistic rationale for RCII (e.g., its role in endothelial dysfunction, macrophage activation).
• Clearly state that this is the first study to apply RCII to ML-based prediction of TVD, distinguishing it from prior stroke-focused applications.

3. Methodological Transparency
Issues:
• Lack of detail on ML model tuning and validation.
• SHAP analysis methods are described but not fully reproducible.
Suggestions:
• Model Tuning: Specify hyperparameter optimization strategy (e.g., grid search, random search) and performance criteria.
• Software Disclosure: List Python/R packages used (e.g., scikit-learn, SHAP, XGBoost).
• SHAP Analysis: Include details on sampling strategy, number of iterations, and whether interaction values were normalized.

4. Model Performance and Validation
Issues:
• Predictive performance is modest (AUC ~0.61).
• RF model shows overfitting; no external validation is performed.
Suggestions:
• Model Enhancement: Consider ensemble stacking or feature engineering (e.g., interaction terms, polynomial features).
• External Validation: Apply the model to a temporally or geographically distinct cohort to assess generalizability.
• Calibration Metrics: Include Brier score, calibration curves, and decision curve analysis to evaluate clinical utility.

5. Data Availability and Transparency
Issues:
• Data is only available upon request; no public repository is used.
• Raw data and code are not linked or described in detail.
Suggestions:
• Deposit anonymized datasets in a repository like Dryad, Figshare, or Zenodo.
• Include a data dictionary and metadata for reproducibility.
• Share model code and SHAP scripts via GitHub or supplementary materials.

6. Figures and Tables
Issues:
• Table 3 lacks visual clarity between training and test sets.
• Figures are referenced but not shown; resolution and labeling are unknown.
Suggestions:
• Table Formatting: Use separate columns or shading to distinguish training vs. test metrics.
• Figure Quality: Ensure all figures have axis labels, legends, and units. Include resolution (DPI) and file format in submission.
• Supplementary Figures: Provide raw SHAP values and interaction matrices as downloadable files.

7. Statistical Analysis
Issues:
• No adjustment for multiple comparisons.
• Some borderline p-values (e.g., monocytes, hypertension) are not discussed.
Suggestions:
• Apply correction methods (e.g., Bonferroni, FDR) where appropriate.
• Discuss borderline predictors in the context of biological plausibility and potential confounding.

8. Interpretability and Clinical Relevance
Issues:
• SHAP results are described but not linked to clinical decision-making.
• No discussion of how RCII could be used in practice.
Suggestions:
• Include a paragraph on how RCII thresholds could inform clinical triage or diagnostic pathways.
• Discuss integration into electronic health records or risk calculators.

9. Limitations and Future Directions
Issues:
• Limitations are acknowledged but not deeply explored.
• No roadmap for prospective validation or implementation.
Suggestions:
• Expand on limitations related to retrospective design, missing data, and regional bias.
• Propose a prospective multicenter study with standardized RCII measurement and ML deployment.

Reviewer 2 ·

Basic reporting

The article is written in professional English, with terminology that is precise and well-suited to the field of cardiovascular medicine. The language is accurate and creates no ambiguity.

Experimental design

The article clearly addresses a relevant clinical question: whether RCII can enhance prediction and early risk stratification for TVD, which is well aligned with the scope of your journal. The authors identify an important gap in the literature regarding the application of this established parameter in stratifying TVD. To evaluate their predictive model, they employed both traditional statistical methods and multiple machine learning algorithms. The study received approval from local ethics committees and was conducted in accordance with the Declaration of Helsinki. The methods and statistical analyses are described thoroughly and with sufficient detail.

Validity of the findings

This is not a replication study; however, the authors acknowledge and build upon previous research in the field. The clinical and laboratory data are comprehensively presented, having been collected through hospital systems for the defined sample size. Clear inclusion and exclusion criteria were applied, a control group was incorporated, and multiple statistical tests were employed. The manuscript also specifies that raw data are available upon request. Collectively, these measures enhance the rigor and credibility of the findings. The conclusions are well connected to the original research and are presented clearly, without overstatement or conflicting interpretations.

Additional comments

- In lines 51–52, the phrase 'significant stenosis (≥50%)' should be amended by omitting 'significant' because current guidelines classify 50–69% stenosis as moderate, not necessarily significant. It would be more accurate to state that triple-vessel disease involves stenoses ≥50% in all three major coronary arteries, recognizing the distinction between moderate and obstructive lesions as per guideline definitions.

- The authors employed both LASSO regression and traditional logistic regression for feature selection and ultimately retained only the four overlapping predictors (Age, Gender, AST, RCII) for ML modeling. While this conservative approach enhances interpretability and stability, it also excludes several variables identified as significant by one method but not the other. Some of these appear clinically relevant, and their omission may limit model robustness. The authors are encouraged to clarify their rationale for prioritizing only the overlapping features and to discuss whether including method-specific predictors or testing their impact in sensitivity analyses might further strengthen the predictive framework.

- The random forest model shows some signs of overfitting, while the MLP appears to generalize better. External validation was not performed, which may limit generalizability. Although this cannot be addressed within the current study, it would be helpful to acknowledge this as a key limitation for the broader applicability of predictive models. I suggest creating a separate paragraph for study limitations, providing sufficient space to properly highlight all gaps and areas for future research.

- Some potential confounding variables, such as the use of medications (e.g., statins, ticagrelor, beta-blockers, antidiabetic drugs, antiplatelets) or patient factors like BMI, are not reported. These factors can significantly influence both lipid and inflammatory markers and represent significant limitations of the study, and should be addressed in a Limitations paragraph.

- The RCII represents a novel parameter, and while large population-based studies have validated its predictive value for outcomes such as mortality and stroke, its role in CAD is less clearly established. There is strong evidence that its components—remnant cholesterol and systemic inflammation—are independently associated with CAD risk, severity, and prognosis. The authors are encouraged to discuss this context, highlighting the rationale for applying RCII to CAD and acknowledging the need for further validation in large CAD-specific cohorts (lines 215-216, “Future prospective, multicenter studies are necessary to validate the clinical utility of RCII in early risk stratification and precision management of complex coronary artery disease”, could be further discussed).

- It would strengthen the discussion if the authors expanded on other emerging biomarkers in the field and clarified why these were not found to be significant in their study (lipoprotein A). This would help highlight the rationale for focusing on RCII.

- It would be helpful if the manuscript discussed how this predictive model enhances clinical interpretability compared to established risk scores, such as the GRACE score.

- It would be valuable for the authors to discuss whether the RCII exhibits differential behavior or levels between the acute and chronic phases of coronary artery disease. Given that TVD typically reflects stable CAD, which may differ pathophysiologically from acute coronary syndromes, clarifying whether RCII dynamics vary across different clinical presentations and stages of CAD.

Overall, the manuscript presents valuable and original research with the potential to advance understanding of the RCII in predicting TVD. However, substantial revisions are required to address methodological clarifications and to enhance the discussion of RCII’s in different CAD phases.

---

## Round 0.2 · accepted · Accept

· Academic Editor

Accept

I can confirm the authors have addressed all the reviewers' comments, and this manuscript is ready for publication.

·

Basic reporting

The manuscript now demonstrates improved clarity and precision in language, with technical terminology appropriately refined. Your revisions to ambiguous phrasing—such as quantifying predictive performance and clarifying the ML framework—enhance readability and interpretability. Figure resolution and table formatting have been addressed, and the inclusion of supplementary SHAP matrices and calibration metrics strengthens transparency. The public deposition of anonymized data aligns with PeerJ’s standards and supports reproducibility.

Experimental design

Your methodological clarifications—particularly around feature selection, hyperparameter tuning, and SHAP analysis—are well-articulated and improve replicability. The rationale for retaining overlapping predictors is clearly explained, balancing interpretability with model stability. Ethical compliance is appropriately documented, and the study design is well-aligned with the clinical question. The expanded discussion on RCII’s mechanistic basis and its relevance to stable CAD adds valuable context, especially in distinguishing its role from acute coronary syndromes.

Validity of the findings

The manuscript acknowledges key limitations, including modest model performance, signs of overfitting in the RF model, and the absence of external validation. Your decision to prioritize RCII interpretability over ensemble optimization is reasonable, though future validation on independent cohorts will be essential to strengthen generalizability. The inclusion of calibration metrics and a dedicated limitations paragraph is commendable. Your discussion of unreported confounders (e.g., medication use, BMI) and emerging biomarkers (e.g., lipoprotein[a]) is thoughtful and appropriately framed within the study’s scope.

Additional comments

Thank you for your comprehensive and collegial revisions. I appreciate the clarity, responsiveness, and scientific rigor with which you addressed the comments from both reviewers. The revised manuscript reflects meaningful improvements across multiple domains and presents a valuable contribution to biomarker-driven coronary risk stratification.
Your clarification that the study focused on stable TVD is appreciated. The acknowledgment that RCII dynamics may differ across CAD phases—and the call for further research in acute settings—adds depth to the clinical relevance. The comparison with GRACE is appropriately contextualized, highlighting RCII’s potential as a laboratory-based complement to existing risk scores. Overall, the manuscript is now more cohesive, transparent, and clinically grounded.

Reviewer 2 ·

Basic reporting

The manuscript is written in clear English and the references are suitable for the topic. The authors are willing to share raw data upon request. The results directly support the hypothesis.

Experimental design

The primary research point is well defined and research question is relevant. Ethical approval has been granted and methods are described sufficiently.

Validity of the findings

The rationale is stated, data are controlled and conclusions are linked to the hypothesis.

Additional comments

I have no further comments.